# DGCFNet: Dual Global Context Fusion Network for remote sensing image semantic segmentation

Yuan Liao[1], Tongchi Zhou[2], Lu Li[3], Jinming Li[3], Jiuhao Shen[3] and Askar Hamdulla[4]

[1] School of Information and Communication Engineering, Zhongyuan University of Technology, Zhengzhou, China
[2] School of Integrated Circuits, Zhongyuan University of Technology, Zhengzhou, China
[3] School of Automation and Electrical Engineering, Zhongyuan University of Technology, Zhengzhou, China
[4] Institute of Information Science and Engineering, Xinjiang University, Urumqi, China



Corresponding author
Tongchi Zhou,
tony2016edu@zut.edu.cn

## ABSTRACT

The semantic segmentation task of remote sensing images often faces various challenges such as complex backgrounds, high inter-class similarity, and significant differences in intra-class visual attributes. Therefore, segmentation models need to capture both rich local information and long-distance contextual information to overcome these challenges. Although convolutional neural networks (CNNs) have strong capabilities in extracting local information, they are limited in establishing long-range dependencies due to the inherent limitations of convolution. While Transformer can extract long-range contextual information through multi-head self attention mechanism, which has significant advantages in capturing global feature dependencies. To achieve high-precision semantic segmentation of remote sensing images, this article proposes a novel remote sensing image semantic segmentation network, named the Dual Global Context Fusion Network (DGCFNet), which is based on an encoder-decoder structure and integrates the advantages of CNN in capturing local information and Transformer in establishing remote contextual information. Specifically, to further enhance the ability of Transformer in modeling global context, a dual-branch global extraction module is proposed, in which the global compensation branch can not only supplement global information but also preserve local information. In addition, to increase the attention to salient regions, a cross-level information interaction module is adopted to enhance the correlation between features at different levels. Finally, to optimize the continuity and consistency of segmentation results, a feature interaction guided module is used to adaptively fuse information from intra layer and inter layer. Extensive experiments on the Vaihingen, Potsdam, and BLU datasets have shown that the proposed DGCFNet method can achieve better segmentation performance, with mIoU reaching 82.20%, 83.84% and 68.87%, respectively.

# INTRODUCTION

With the development of aerospace technology and sensor technology, researchers can easily obtain high-resolution remote sensing images with rich spatial details and potential semantic content (*Jiang, Jiang & Jiang, 2020*; *Hoeser & Kuenzer, 2020*), including various geographic spatial objects such as buildings, cars, roads, bare soil, *etc.* Remote sensing image semantic segmentation is the pixel-level classification task of the entire image, which marks the category of each pixel in the image. It can provide technical support for applications of urban planning (*Yan et al., 2019*; *Bi et al., 2020*; *Huang et al., 2021*), agricultural production (*Liu et al., 2019*; *Sheikh et al., 2020*), disaster monitoring (*Schumann et al., 2018*), land change (*Samie et al., 2020*), road extraction (*Lian et al., 2020*), *etc.* However, due to the inherent characteristics of high-resolution remote sensing images, remote sensing target segmentation tasks mainly face three challenges: (1) Complex background: multiple terrain features are mixed together in the remote sensing image, forming a complex background that makes accurate target segmentation extremely difficult; (2) high inter-class similarity: objects of different categories may have similar appearance features in remote sensing images, which significantly increases the difficulty of distinguishing different categories; (3) large intra-class variance: there are significant differences in buildings at different locations, targets of the same category may exhibit significant internal differences in high-resolution remote sensing images, making the recognition of them more complex. Traditional semantic segmentation methods such as support vector machines (*Guo, Jia & Paull, 2018*), random forests (*Pal, 2005*), and conditional random fields (*Krähenbühl & Koltun, 2011*) have achieved good results in recent years, but their generalization is poor and they can no longer meet the current demand for high-precision segmentation.

With the rapid development of deep learning technology, convolutional neural networks (CNNs) have been widely used as an important feature extraction method in various computer vision tasks, especially in image semantic segmentation. *Long, Shelhamer & Darrell (2015)* replaced traditional fully connected layers with convolutional layers and proposed a fully convolutional network (FCN) that achieved pixel level segmentation for the first time. Inspired by this, *Ronneberger, Fischer & Brox (2015)* proposed a network based on encoder and decoder structures, called UNet. The encoder is used to extract features, while the decoder combines high-level semantic and low-level spatial information to restore the resolution of the image, and the missing feature information due to downsampling is compensated through skip connections to further improve segmentation accuracy. However, pixel-level fine-grained classification tasks in semantic segmentation require rich high-level semantic information, and UNet just simply concatenates low-level features with high-level features, making it difficult to obtain satisfactory segmentation results. Although subsequent researchers have attempted to modify the structure through various techniques, such as the expansion of receptive fields (*Zhao et al., 2017*; *Chen et al., 2018a, 2017, 2018b*), extraction of more contextual information (*Zhou et al., 2019*; *He et al., 2019*; *Yu et al., 2020*; *Yuan, Chen & Wang, 2020*), and attention mechanisms (*Hu, Shen & Sun, 2018*; *Zhao et al., 2021*), these CNN-based

methods still cannot directly capture long-distance contextual relationships, resulting in room for improvement in segmentation accuracy.

Inspired by the tremendous success of Transformer (*Vaswani et al., 2017*) in natural language processing (NLP), researchers have applied it to computer vision tasks such as semantic segmentation. *Dosovitskiy (2020)* proposed the Vision Transformer (ViT) model, which divides an image into multiple small image blocks and linearly embeds them into a standard Transformer to achieve image classification. *Zheng et al. (2021)* proposed Segmentation Transformer (SETR) using ViT as the backbone network, demonstrating for the first time the feasibility of Transformer in semantic segmentation. However, the computational complexity of the multi-head attention mechanism in Transformer is proportional to the square of the input sequence length, which limits its application in high-resolution images. To address this issue, *Liu et al. (2021)* proposed a Swin Transformer based on shift windows, which adopts self-attention mechanisms within local windows. *Wang et al. (2021b)* proposed a pyramid visual Transformer, which replaces traditional multi-head attention (MHA) with spatial-reduction attention (SRA) to reduce the number of key-value pairs, thus reducing computational complexity. Although Transformer-based methods utilize multiple kinds of self-attention mechanisms to model long-distance dependencies, there are certain limitations in capturing local features and maintaining scale invariance (*Xu et al., 2021*). On the contrary, CNN has favorable locality and scale invariance by using convolutional kernels to calculate the interrelationships between adjacent pixels. Therefore, we can conclude that CNN and Transformer focus on different aspects in semantic segmentation respectively. On the one hand, CNN can better extract local detail information, but its limited receptive field limits its ability to model long-range contextual dependencies. On the other hand, Transformer is effective at establishing global dependencies, but cannot extract local specificity well.

From the above analysis, combining CNN and Transformer can make up for their shortcomings and enhance the advantages of both. To this end, *Chen et al. (2021)* proposed TransUNet, which uses the CNN structure of UNet for feature extraction, and then utilizes the self-attention mechanism of Transformer to globally model the features, achieving excellent performance in medical image segmentation. CNN and Transformer are integrated into a single architecture, called Defect Transformer (*Wang et al., 2023*), DefT to encode local positions through the locally position-aware block and model multi-scale global contextual relationships by the lightweight multi-pooling self-attention block, achieving high-precision defect segmentation results. *He et al. (2022)* proposed a dual encoder parallel structure ST-UNet based on CNN and Swin Transformer, in which the relationship aggregation module can fuse the global and local features obtained by the main encoder and auxiliary encoder. *Ding et al. (2021)* proposed WiCoNet to extract local features through two branches of convolutional neural networks, and then fuse the two branches of features using Transformer. LEFormer (*Chen et al., 2024*) introduced a hybrid architecture that integrated CNN and Transformer components for precise lake extraction from remote sensing images. It employed a cross-encoder fusion module to merge the local spatial details captured by the CNN encoder with the global features derived from the Transformer encoder, thereby enhancing the precision of mask prediction. It can be seen

that the above segmentation models only use Transformer in the encoder, and there is little research on the role of Transformer in the decoder. However, UNetFormer (*Wang et al., 2022b*) proposed a hybrid architecture of CNN and Transformer in series, which adopted ResNet18 as the encoder, and designed a global-local transformer block (GLTB) to build the decoder, and finally achieved accurate segmentation results for remote sensing images. CMTFNet (*Wu et al., 2023*) presented a novel semantic segmentation network with a CNN encoder and a multi-scale Transformer decoder architecture. It constructed a Transformer decoder based on the multiscale multihead self-attention (M2SA) module, which efficiently extracted and fused local and multi-scale global context information from remote sensing images.

UnetFormer and CMTFNet achieved remarkable results, attributed to their unique and efficient network architectures. Consequently, we opted to adopt a network architecture that employs a CNN as the encoder and a Transformer as the decoder. Although UnetFormer utilized Transformers to acquire global information, it did not fully capitalize on the feature information between different layers. Therefore, we have repeatedly used various methods to obtain global context information in order to further enhance the model's performance.

Based on the above analysis, a Dual Global Context Fusion Network (DGCFNet) with CNN as the encoder and Transformer as the decoder is proposed for high-precision semantic segmentation of remote sensing images. In the encoding phase, the ResNet-18 model is used to extract multi-scale features. In the decoding part, to improve the recognition ability of object boundaries and details, a dual-branch global extraction module (DGEM) composed of global attention branch and global compensation branch is proposed to fuse local information and contextual information. Meanwhile, a cross-level information interaction module (CIIM) based on Transformer is proposed to enhance the contextual correlation between different levels. Finally, to enhance the representation ability of image semantic content, the feature interaction guided module (FIGM) is proposed, which adaptively fuses the intra-layer global context relationship with the inter-layer context relationship.

The main contributions of this work are as follows:

(1) A novel network specifically designed for semantic segmentation tasks in remote sensing images is proposed. This network integrates CNN and Transformer, which can better model long-range dependencies while preserving spatial features.

(2) A dual-branch global extraction module is proposed, which consists of a global attention branch based on multi-head attention mechanism and a global compensation branch containing multiple average pooling layers to extract rich global contextual information and enhance local information.

(3) A cross-level information interaction module based on Transformer is proposed to fuse two information from different levels, thus enhancing recognition ability at different scales.

(4) A feature interaction guided module is proposed, which can effectively integrate inter-layer contextual information and intra-layer contextual information to further refine segmentation results.

# RELATED WORK

## CNN-based semantic segmentation methods

FCN (*Long, Shelhamer & Darrell, 2015*) is a CNN structure that solves semantic segmentation problems in an end-to-end manner. It uses transposed convolutional layers instead of fully connected layers in traditional image classification networks, and can gradually reduce the resolution of feature maps through stacked convolutional layers to obtain more semantic information. Afterwards, FCN-based methods dominated in the field of semantic segmentation of remote sensing images. However, due to the simplicity of its decoder, it is prone to misclassify image pixels, resulting in low segmentation accuracy. To address this issue, UNet (*Ronneberger, Fischer & Brox, 2015*) proposed an encoder-decoder network that utilizes skip connections to preserve the spatial information of the input image, allowing UNet to utilize semantic information at different levels while maintaining high-resolution features. Therefore, nowadays, the encoder-decoder structure has become the mainstream network for remote sensing images semantic segmentation. Nevertheless, due to the limitations of convolutional kernels, the limited receptive field can only extract local semantic features and lack the ability to global modeling of the entire image. To solve this problem, PSPNet (*Zhao et al., 2017*) constructed a feature pyramid through multiple parallel convolutional and pooling layers, which can receive receptive fields of different sizes to capture multi-scale contextual information. DeeplabV3 (*Chen et al., 2018a*) introduced an atrous spatial pyramid pooling module that captures multi-scale contextual information through multiple dilated convolutions with different dilation rates. In the field of remote sensing images, $A^2$-FPN (*Li et al., 2022*) proposed an attention aggregation module that adaptively selects features at different scales through attention mechanisms, and combines feature pyramid networks to achieve effective learning and utilization of multi-scale features. MAResU-Net (*Li et al., 2021a*) proposed a linear attention mechanism and incorporated it into the skip connections of the encoder decoder structure to make the network automatically focus on important regions of the image, thereby improving segmentation accuracy. LANet (*Ding, Tang & Bruzzone, 2020*) proposed a local attention mechanism that introduces a local perception module to perceive the target scale changes, meanwhile, utilizes multi-scale feature fusion and fine-grained semantic segmentation techniques to improve segmentation accuracy and robustness. *Li et al. (2021b)* introduced multiple attention mechanisms to simultaneously focus on information from different aspects and scales of an image. Although the above methods have achieved significant segmentation results, they mainly obtain global information by aggregating local features extracted by convolutional kernels, rather than directly modeling global relationships. To address the issue of image feature loss, *Wang (2023)* introduced a new segmentation network, G-Lite-DeepLabV3. By combining group convolution with an attention mechanism, their model effectively handles complex semantic features and enhances the network's response to important features. In order to distinguish between benign and malignant tissues in medical images, *Damkliang et al. (2023)* proposed an integrated method that combines a UNet-based architecture with attention gate units and residual convolutional units for prostate cancer (PCa) tissue

analysis, achieving excellent results. *Dong et al. (2024)* proposed the Adaptive Adjacent Context Negotiation Network (A$^2$CN-Net) to address issues related to large-scale changes and small-sized targets. This network utilizes a composite fast Fourier convolution module to enhance global context information and adaptively fuses multi-level features. *Qu et al. (2024)* proposed a deep multi-branch residual Unet with fused inverse weighting gated control. By introducing a deep multi-branch residual module that utilized convolution, detailed image features were extracted at deeper levels, the inverse weighting gated control module enhanced the diversity of upsampling information through counterclockwise horizontal transmission attention. Finally, at the highest level of the U-shaped encoder, a pyramid attention mechanism with different receptive fields was used to address the loss of pixel-level information leading to boundaries. Unlike these networks, our proposed model not only uses convolutional neural networks but also incorporates Transformers to further capture the global context information.

## Transformer-based semantic segmentation methods

Transformer (*Vaswani et al., 2017*) was initially applied in the field of NLP. To apply it to image semantic segmentation, the image processing task is transformed into the sequence task of 1D image blocks. In this way, contextual information can be obtained through the powerful sequence-to-sequence modeling ability of Transformer structure. Most Transformer-based semantic segmentation networks still follow the encoder-decoder structure and can be roughly divided into two categories. The first category is pure Transformer methods. For example, SETR (*Dosovitskiy, 2020*) utilized the self-attention mechanism in Transformer to capture the dependency relationships between pixels in an image, thereby significantly improving the performance of semantic segmentation. However, the length of the input sequence is linearly related to the image resolution, which can lead to low efficiency in processing high-resolution images. Swin Transformer (*Liu et al., 2021*) proposed a window self-attention mechanism that divides an image into a series of small image blocks and only calculates the self-attention of the image region within the window. The Transformer structure has also demonstrated strong performance in remote sensing images. CGVT (*Deng et al., 2023*) used the crisscross attention mechanism to extract features with different scales at different levels. By combining the crisscross attention mechanism with the ViT model, it can improve the understanding of details and global structures, thus enhancing the accuracy of semantic segmentation. DCSwin (*Wang et al., 2022a*) adopted Swin Transformer as the encoder to extract contextual information and proposed a novel densely connected feature aggregation module and two attention blocks to connect four hierarchical Transformer features, thereby obtaining rich multi-scale information and context relationships. Another type is a hybrid network that combines the advantages of CNN and Transformer. For example, ST-UNet (*He et al., 2022*) constructed a dual encoder structure in parallel with Swin Transformer and CNN, and hierarchically integrated the global dependencies of Swin Transformer into the features of CNN through a relational aggregation module. *Ding et al. (2021)* proposed a wide context network that uses an adaptive perceptual context feature fusion mechanism to dynamically select and fuse features of different scales. BANet

(*Wang et al., 2021a*) proposed a bilateral perception network that includes dependency paths and texture paths. The former extracted global contextual relationships through Transformer, while the latter extracted texture features through CNN. AerialFormer (*Hanyu et al., 2024*) employed a Transformer as the encoder and utilized multi-dilated convolutional neural networks (MD-CNNs) for the decoder. By considering both local and global context information, it achieved powerful representation and enabled high-resolution segmentation. Distinguished from these networks, our proposed model builds upon the use of Transformer and further investigates the integration of information across different layers by exploiting their inter-layer relationships, thereby enhancing the model's segmentation performance.

# METHODOLOGY

## Overall structure

As shown in Fig. 1, the proposed DGCFNet network consists of an encoder and a decoder. The encoder uses ResNet18 as the backbone network to extract multi-scale semantic features. It consists of four stages of Resblocks, and each stage comprises two $3 \times 3$ convolution layers. Except for the first stage that uses a max pooling layer, the other three stages change the resolution of the feature map through a $3 \times 3$ convolution with a stride of 2. Therefore, given an image with size of $H \times W$ four different sizes of feature maps are generated: $H/4 \times W/4$, $H/8 \times W/8$, $H/16 \times W/16$, $H/32 \times W/32$. The decoder includes a multi-scale Transformer block that obtains multi-scale contextual information, two redistribution weight blocks that efficiently fuse the obtained intra-layer and inter-layer information, and a Seghead that achieves pixel-level classification predictions. Specifically, in the decoder, a $1 \times 1$ convolution is first used to unify the channel dimensions of the four output feature maps of the encoder to 64, and then the corresponding feature images of the decoder are fused through skip connections. In this way, the fused features include local semantic features generated by the encoder and global semantic features generated by the decoder, which is beneficial for high-precision semantic segmentation of remote sensing images. At the same time, the feature map obtained from the last ResBlock is input into the multi-scale Transformer block, which undergoes a $3 \times 3$ convolution and a DGEM module to extract multiscale contextual information. In addition, inter-layer information and intra-layer information are fused through two redistribution weight modules, which is composed of a DGE module, a CII module, and a FIG module in sequence. The details of the proposed DGEM, CIIM and FIGM are following.

## Dual-branch global extraction module

For complex remote sensing images, both local and global contextual information are crucial for semantic segmentation. Therefore, a dual-branch global extraction module is proposed to extract detailed local information and rich global information. which includes a global attention branch (GAB) and a global compensation branch (GCB). As shown in Fig. 2, the global attention branch uses a multi-head self-attention (MSA) with windows to capture contextual information, while the global compensation branch aggregates

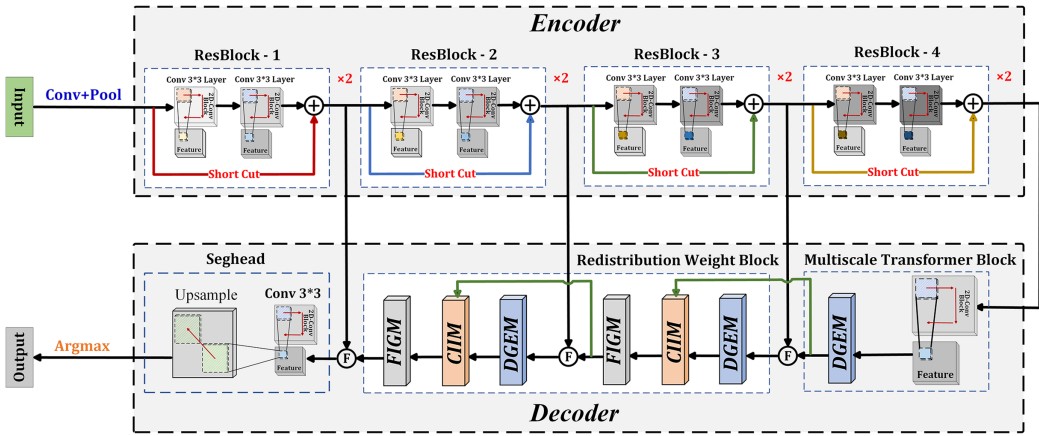

**Figure 1 Overall network architecture of the proposed DGCFNet.**

contextual information and local information extracted from multiple average pooling layers with different sizes.

Inspired by *Wang et al. (2022b)*, the global attention branch first utilizes $1 \times 1$ convolution to expand the channel dimension of the input feature map by three times. Then, the cross window context interaction module (CWCI) is used to divide the windows into four parts: upper left, upper right, lower left, and lower right. The horizontal average pooling layer is used to establish the horizontal relationship between the windows, and the vertical average pooling layer is used to establish the vertical relationship between the windows. Based on this cross window pixel dependency, the relationship between any two windows can be established and then the global contextual relationships of image can be obtained. Specifically, given an input feature $X \in \mathbb{R}^{C \times H \times W}$, where $C$ represents the channel dimension, $H$ and $W$ represents the height and width of the feature map, the output of global attention branch $X_{cwci}$ can be represented as:

$$X_1 = \text{Conv}_{1\times1}(X) \tag{1}$$
$$X_{cwci} = X_1 + CWCI(\text{MSA}(\text{LN}(X_1))) \tag{2}$$

where LN means layer normalization.

Meanwhile, to extract detailed local information, the global compensation branch consists of multiple average pooling layers with different sizes in parallel. Specifically, three average pooling layers $\text{AvgPool}_{(5,2)}$, $\text{AvgPool}_{(9,4)}$, $\text{AvgPool}_{(17,8)}$ and one global average pooling layer are first used to extract multi-scale contextual information, where $\text{AvgPool}_{(i,j)}$ represents an average pooling with size of i × i and stride of j. Then, upsampling operations are adopted to restore the four outputs to their original input size. The information from each sub-branch are also fuse with the original input information through summation operation. At the same time, in each branch, a $1 \times 1$ convolution is used to reduce computational complexity, and a $3 \times 3$ convolution is used to further feature learning. After information fusion, a $1 \times 1$ convolution is used to restore the original channel dimension, and residual connection is employed to preserve the input

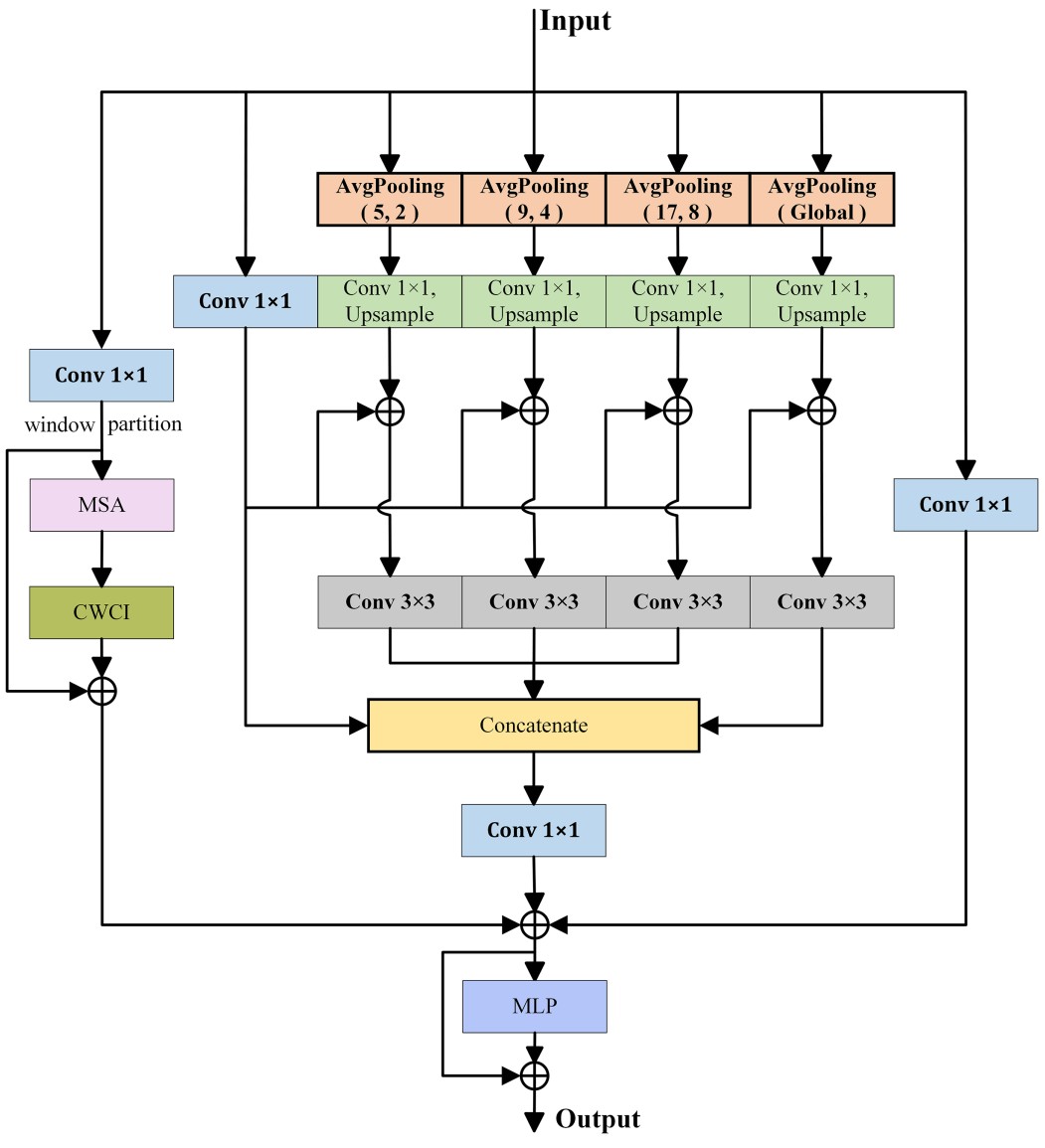

**Figure 2  Illustration of DGEM.**         

feature information. Finally, the representation capability can be further increased through a multilayer perceptron (MLP). In summary, the output of DGEM $X_{out}$ can be expressed as:

$$X_{2,7} = \text{Conv}_{1\times1}(X) \tag{3}$$

$$X_{3,4,5,6} = \text{Conv}_{3\times3}\big(X_2 \oplus \text{Upsample}\big(\text{Conv}_{1\times1}\big(\text{Avgpool}_{(i,j)}(X)\big)\big)\big) \tag{4}$$

$$X_{out1} = \text{Conv}_{1\times1}(\text{Concat}(X_2, X_3, X_4, X_5, X_6)) \oplus X_7 \oplus X_{\text{cwci}} \tag{5}$$

$$X_{out} = X_{out1} + \text{MLP}(\text{LN}(X_{out1})) \tag{6}$$

where $X_{2-7}$ represents the output of different sub-branches in the global compensation branch, and $\oplus$ represents the element-wise addition operation. Upsample denotes upsampling operation.

## Cross-level information interaction module

In semantic segmentation, the feature maps at different stages contain feature information at different levels. Typically, only a few simple and similar modules are used within each stage to process features to obtain the local interaction However, considering local enhancement only is not enough. It is necessary to learn cross layer contextual interactions on adjacent layer features, and thus CIIM module is proposed to explore the impact of correlations between features at different levels on semantic segmentation.

In semantic segmentation, we conclude that the element-wise multiplication of two feature maps contains common information, which can highlight prominent features more. While the element-wise addition of the two feature maps allows information to be complementary without missing unique information, which is beneficial for refining objects. Based on this observation, the proposed CIIM module is shown in Fig. 3 and consists of two self-attention operations. Given inputs $f_1$ and $f_2$, they are resized to $f_1, f_2 \in \mathbb{R}^{c \times h \times w}$ firstly by the bilinear interpolation. Then, the element-wise multiplication and addition of these two feature maps are represented as $f_m = f_1 \otimes f_2$ and $f_s = f_1 \oplus f_2$, respectively, where $\otimes$ represents the element-wise multiplication operation. In addition, to reduce computational costs, the channel dimensions of them are halved through $1 \times 1$ convolution, resulting in two new features $f_Q \in \mathbb{R}^{(c/2) \times h \times w}$ and $f_K \in \mathbb{R}^{(c/2) \times h \times w}$. Consequently, the self-attention weight can be expressed as:

$$C = \text{Softmax}(f_Q \circledast f_K) \tag{7}$$

where Softmax $(\cdot)$ is the softmax activation function, and $\circledast$ is matrix multiplication. In this way, the interdependence relationship between the coexistence information of $f_m$ and the comprehensive information of $f_s$ can be established.

Next, the generated self-attention weight matrix are multiplied with $f_{V1}$ and $f_{V2}$ to obtain two new feature maps $f_1^* \in \mathbb{R}^{c \times h \times w}$ and $f_2^* \in \mathbb{R}^{c \times h \times w}$, where $f_{V1} = \text{conv}_{1 \times 1} \circledast f_1$ and $f_{V2} = \text{conv}_{1 \times 1} \circledast f_2$ are both linear mappings of input $f_1, f_2$, Afterward, $f_1^*$ and $f_1$ are added through residual connections, and the same is done for $f_2^*$ and $f_2$ Finally, the fused feature $f_{12} \in \mathbb{R}^{c \times h \times w}$ is obtained through convolution and element-wise summation. Specifically, the calculation process of $f_{12}$ is:

$$f_1^* \in f_{V1} \circledast C \tag{8}$$

$$f_1^* \in f_{V2} \circledast C \tag{9}$$

$$f_{12} = \text{Conv}_{1 \times 1}\left(\text{Conv}_{3 \times 3}\left(f_1^* \oplus f_1\right) \oplus \text{Conv}_{3 \times 3}\left(f_2^* \oplus f_2\right)\right) \tag{10}$$

The output of CIIM module $f_{12}$ inherits the properties of two adjacent input feature maps $f_1$ and $f_2$, which can further supplement the local enhanced features and is beneficial for simultaneously characterizing and recognizing different objects.

## Feature interaction guided module

In the previous section, the dual branch global extraction module uses self-attention mechanism to capture the correlation between different pixels, which can capture long-distance relationships and global contextual relationships in the image. The cross

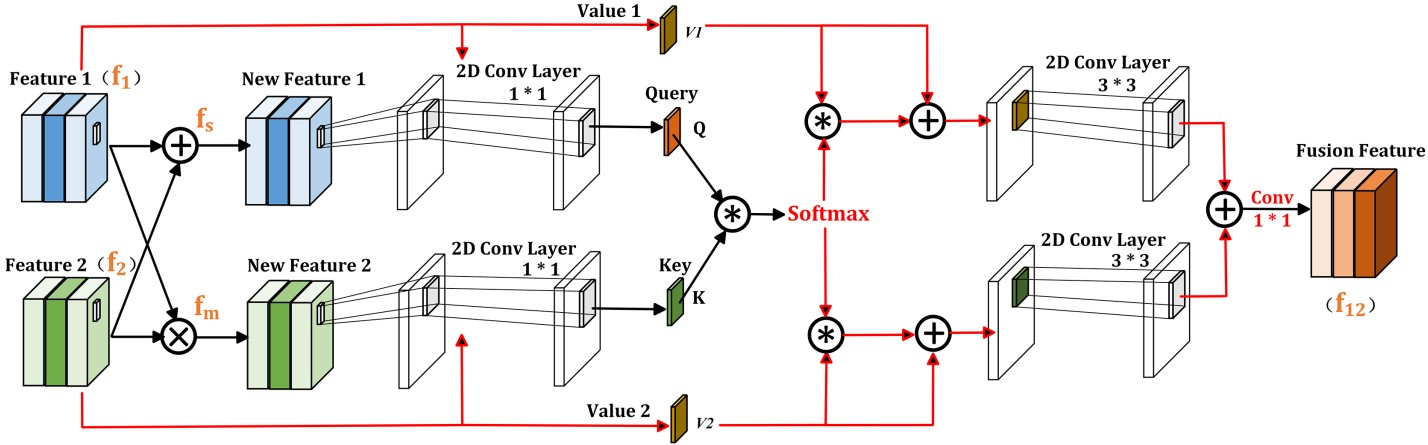

**Figure 3  Illustration of CIIM.**

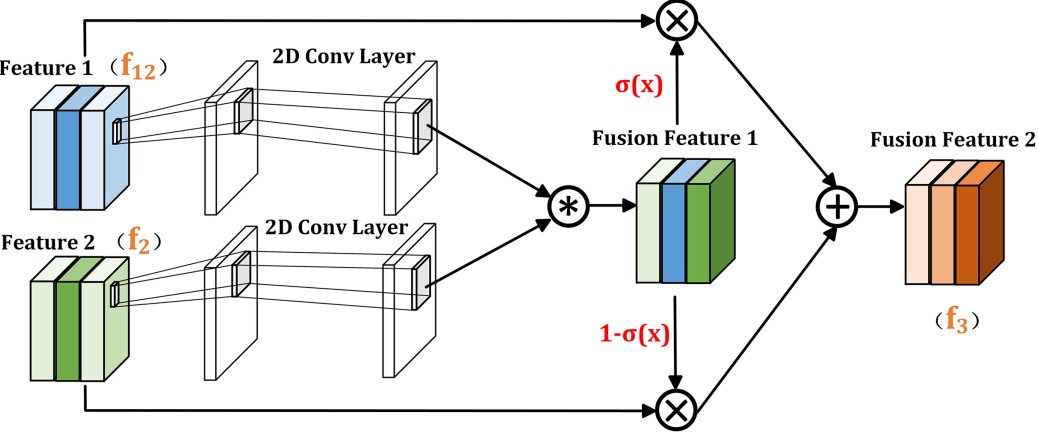

**Figure 4  Illustration of FIGM.**

layer information interaction module uses cross self-attention to establish correlations between different levels, which can better understand the relationship between different scales and semantic features, and improve the processing ability of details and local information. To further promote the fusion of output information from the above two modules and improve feature representation capabilities, the FIGM shown in Fig. 4 is proposed.

Given that the two inputs of FIGM module are $f_{12}$ (output of the CIIM module) and $f_2$ (output of the DGEM module), and the feature vectors of their corresponding pixels are $\vec{v_1}$ and $\vec{v_2}$ respectively. The output of FIGM $\vec{f_3}$ can be expressed as:

$$\sigma = \text{Sigmoid}((w_1(\vec{v_1})) \cdot ((w_2(\vec{v_2})))) \tag{11}$$

$$\vec{f_3} = \sigma\vec{v_1} + (1 - \sigma)\vec{v_2} \tag{12}$$

where Sigmoid (·) represents the Sigmoid activation function, "·" is a placeholder, used to indicate that the input to the Sigmoid function can be any real number. $w_1$ and $w_2$ represent the weights generated by $1 \times 1$ convolutions.

# DATASET AND EXPERIMENTAL SETTING

## Datasets

Vaihingen dataset (*ISPRS, 2019b*): The vaihingen dataset is a regional dataset collected in Germany urban scenes. It consists of 33 high-resolution remote sensing images captured by drones, with an average size of 2,494 × 2,064 pixels. Each image in this dataset has a corresponding ground truth label, which includes six categories: The white, blue, cyan, green, gold, and red in the figure represent impressive surface, building, low vegetation, tree, car, clutter respectively. In this article, 16 images are selected for training, and the remaining images are used for testing. Each image is segmented into small blocks of 1,024 × 1,024 pixels to meet the experimental requirements. Similar to *Li et al. (2022)*, *Ding, Tang & Bruzzone (2020)*, the Vaihingen dataset only has a small number of clutter/background, so the results of clutter/background are not presented.

Potsdam dataset (*ISPRS, 2019a*): The Potsdam dataset is an aerial image dataset collected in the Potsdam urban scene in Germany, widely used in computer vision and remote sensing image processing. It contains 38 high-resolution remote sensing images, each with a size of 6,000 × 6,000 pixels. Similar to the Vaihingen dataset, it also contains six identical categories and doesn't need to show the results of clutter/background. Meanwhile, each image is labeled with pixel-wise ground-truth annotation. We randomly select 22 images for training, two for validation, and 14 for testing. To meet the experimental requirements, the original image is cropped into small image blocks of 1,024 × 1,024 pixels.

BLU dataset (*Ding et al., 2021*): The BLU dataset is captured in Beijing by 21st Century Aerospace Technology Corporation using the Beijing-2 satellite. It contains 4 RGB optical remote sensing images, each with a size of 15,680 × 15,680 pixels. Each original image is overlapped and divided into 64 small images of 2,048 × 2,048 pixels. They also have the corresponding ground-truth labels and include six categories: background/barren, built-up, vegetation, water, Farmland, and road. We randomly select 196 images for training, 28 for validation, and 32 for testing.

## Implementation details

Training settings: All experiments in this article are implemented using the PyTorch framework on NVIDIA Tesla A800 GPU. We adopt AdamW optimizer to optimize model parameters, set the initial learning rate to 6e−4, and adjust the learning rate using cosine strategy. For the Vaihingen and Potsdam datasets, the images are further randomly cropped into small blocks of 512 × 512, meanwhile, data augmentation is performed using random scaling and flipping techniques, with a batch size set to 16. For the BLU dataset, the images are randomly cropped into small blocks of 256 × 256, and data augmentation is also performed using random flipping and random cropping operations, with a batch size set to 32.

Loss function: To alleviate the problem of class imbalance caused by large differences in the number of targets in remote sensing images, the loss function used in this article is a

combination of cross entropy loss $L_{ce}$ and dice loss $L_{dice}$ (*Deng et al., 2023*). The specific formula is as follows.

$$L = L_{ce} + L_{dice} \tag{13}$$

$$L_{ce} = -\sum_{i=1}^{n} t_i \log(p_i) \tag{14}$$

$$L_{dice} = 1 - \sum_{i=1}^{n} \frac{2t_i p_i}{t_i + p_i} \tag{15}$$

where $t_i$ represents the ground truth, and $p_i$ is the maximum probability of softmax for class i.

Evaluation metrics: To fairly compare the performance of various models, several commonly used metrics (*Marmanis et al., 2018*; *Mou, Hua & Zhu, 2019*) are adopted, including overall accuracy (OA), mean intersection over union (mIoU), and mean F1-score (mF1). Their calculation formulas are as follows:

$$Precision = TP/(TP + FP) \tag{16}$$

$$Recall = TP/(TP + FN) \tag{17}$$

$$F1 = 2 \times \frac{Precision \times Recall}{Precision + Recall} \tag{18}$$

$$IoU = \frac{TP}{TP + FP + FN} \tag{19}$$

$$OA = \frac{TP + TN}{TP + TN + FP + FN} \tag{20}$$

where TP, FP, TN, and FN represent true positive, false positive, true negative, and false negative, respectively. OA represents the ratio of correctly predicted pixels to the total number of pixels. IoU is defined as the intersection and union ratio of predicted and ground-truth maps, and the average IoU of all categories is mIoU. The F1 score is the harmonic mean of Precision and Recall for each category.

# EXPERIMENTAL RESULTS AND ANALYSIS

## Ablation study

To evaluate the effectiveness of each module in the proposed network, ablation studies are conducted on the Potsdam dataset in this section. The baseline model adopts U-Net network with ResNet18 as the encoder, and the decoder only uses convolution and upsampling interpolation operations.

(1) Effect of DGEM module: As shown in Table 1, after introducing the Transformer based DGEM module in the decoding section, the values of mean F1, OA, and mIoU increase by 0.33%, 0.57%, and 0.43%, respectively. Especially in the Building category, the F1 value increase by 1.00%, which proves the effectiveness of DGEM. Moreover, to further explore the necessity of global attention branch and global compensation branch in

**Table 1  Ablation study of each component of the DGCFNet (%).**

| Architecture | Imp.surf. | Building | Lowveg. | Tree | Car | Mean F1 | OA | mIoU |
|---|---|---|---|---|---|---|---|---|
| Baseline | 92.06 | 94.19 | 85.51 | 86.39 | 94.04 | 90.44 | 89.03 | 82.76 |
| Baseline+DGEM | 92.69 | 95.19 | 85.84 | 86.36 | 93.76 | 90.77 | 89.46 | 83.33 |
| Baseline+DGEM+CIIM | 92.55 | 95.14 | 85.94 | 86.46 | 93.88 | 90.79 | 89.45 | 83.36 |
| Baseline+DGEM+CIIM+FIGM | **92.90** | **95.35** | **86.33** | **86.67** | **94.14** | **91.08** | **89.80** | **83.84** |

**Note:**
Values highlighted in bold are the top performance. The middle area of the table corresponds to the F1 values for all categories.

DGEM, the segmentation performance is further evaluated when the global attention branch GAB and global compensation branch GCB are used separately. As reported in Table 2, removing only the global compensation branch decrease the value of mIoU by 0.32%, while removing the global attention branch decrease by 0.30%. However, when both GAB and GCB are added simultaneously, the F1 values for Imp.surf. and Car decrease slightly compared to when only GAB is added. This is because GCB tends to favor more long-range context information, while GAB can obtain more detailed information. Consequently, the detailed information for Imp.surf. and Car is somewhat interfered with by GCB. The above results confirm that both branches are crucial.

(2) Effect of CIIM module: The addition of the CIIM model enables the model to simultaneously utilize contextual information from the current and adjacent layers. From Table 1, it can be seen that the CIIM model can improve F1 values for most categories. Meanwhile, mean F1 and mIoU values also show some improvement, reaching 90.79% and 83.36%, respectively. However, the results of Imp.surf. and Building have indeed declined, because we simultaneously incorporate the DGEM and CIIM modules with equal weights, this results in better outcomes in DGEM or CIIM needing to be equally weighted with the opponent, which notably causes information interference and limits the model's performance. To further investigate the effectiveness of CIIM in different locations, the second and third DGEM are selected for ablation studies. As shown in Table 3, when the CIIM after the third DGEM is removed, mIoU value decreases by 0.16%. When removing the other one, mIoU value decreases by 0.21%. However, when both $f_{12}$ and $f_{23}$ are added simultaneously, the F1 scores for Tree and Car decrease slightly, indicating that information from different features can interfere with each other, leading to a decline in performance for certain categories. Nevertheless, when both $f_{12}$ and $f_{23}$ are added simultaneously, the F1 scores for Tree and Car decrease slightly, indicating that information from different features can interfere with each other, leading to a decline in performance for certain categories. The above results indicate the necessity of using two CIIM modules simultaneously.

(3) Effect of FIGM module: From Table 1, it can be observed that after adding the FIGM module, our DGCFNet model achieve the best results in all categories. Especially for the Building category, the final model improve by 1.16% compared to the baseline. Conversely, not using the FIGM module may result in a 0.35% reduction in mIoU, as these weights may be incorrectly assigned through pixel-wise addition. This indicates that the addition of

**Table 2 Results of ablation studies on the Potsdam dataset for the DGEM module.**

| Method | Imp.surf. | Building | Lowveg. | Tree | Car | Mean F1 | OA | mIoU |
|---|---|---|---|---|---|---|---|---|
| GAB | **93.02** | 94.93 | 85.38 | 86.33 | **94.25** | 90.78 | 89.48 | 83.37 |
| GCB | 92.56 | 95.02 | 86.14 | 86.47 | 93.98 | 90.83 | 89.50 | 83.42 |
| GAB+GCB | 92.90 | **95.35** | **86.33** | **86.67** | 94.14 | **91.08** | **89.80** | **83.84** |

Note:
Values highlighted in bold are the top performance.

**Table 3 Results of ablation studies on the Potsdam dataset for the CIIM module.**

| Method | Imp.surf. | Building | Lowveg. | Tree | Car | Mean F1 | OA | mIoU |
|---|---|---|---|---|---|---|---|---|
| CIIM ($f_{12}$) | 92.65 | 95.23 | 86.16 | 86.69 | **94.15** | 90.98 | 89.64 | 83.67 |
| CIIM ($f_{23}$) | 92.60 | 95.21 | 86.11 | **86.72** | 94.06 | 90.94 | 89.59 | 83.61 |
| CIIM ($f_{12}$)+CIIM ($f_{23}$) | **92.90** | **95.35** | **86.33** | 86.67 | 94.14 | **91.08** | **89.80** | **83.84** |

Note:
Values highlighted in bold are the top performance.

feature maps may lead to information confusion, making it difficult to correctly learn specific semantic regions and boundaries.

The Baseline model will cause large areas of Imperfect Surfaces to be incorrectly segmented into Buildings. The addition of DGEM module significantly reduces erroneous segmentation, but inaccurate boundary segmentation is caused by only using multi-scale contextual information within the current layer. Then, by introducing inter-layer contextual information through the CIIM module, the boundary segmentation results are further improved. However, the fusion of contextual information from the current and adjacent layers through simple element-wise addition may lose spatial information, making it difficult for the model to accurately locate the boundaries of objects during prediction. By incorporating the FIGM module, the output boundaries of the proposed DGCFNet model become smoother and there is a significant reduction in erroneous segmentation.

## Comparison with other methods

This section will compare the proposed DGCFNet with some existing methods, including FCN (*Long, Shelhamer & Darrell, 2015*), LANet (*Ding, Tang & Bruzzone, 2020*), A$^2$-FPN (*Li et al., 2022*), MANet (*Li et al., 2021b*), MAResUNet (*Li et al., 2021a*), BANet (*Wang et al., 2021a*) and DCSwin (*Wang et al., 2022a*). Except for the BANet and DCSwin are methods based on Transformer structure, the others are all based on CNN. As with the experiments involving A$^2$-FPN, BANet, MANet, we crop the original images and perform data augmentation by rotating, resizing, horizontally flipping, vertically flipping, and adding random noise.

(1) Results on Vaihingen dataset: The quantitative results of the comparative methods are shown in Table 4. We can see that the proposed method can achieve the highest values of meanF1, OA, mIoU, and the second highest F1 value. Compared with BANet, the meanF1 value of our method increases by 1.42%, OA increases by 0.70%, and mIoU

**Table 4 Quantitative Comparison with state-of-the-art models on the Vaihingen dataset (%).**

| Method | Imp.surf. | Building | Lowveg. | Tree | Car | Mean F1 | OA | mIoU |
|---|---|---|---|---|---|---|---|---|
| FCN | 95.77 | 93.74 | 82.94 | 88.61 | 66.37 | 85.49 | 91.70 | 76.03 |
| BANet | 96.21 | 93.49 | 83.53 | 89.43 | 80.18 | 88.57 | 92.20 | 80.00 |
| LANet | 96.44 | 94.51 | 84.26 | 89.60 | 82.22 | 89.41 | 92.69 | 81.30 |
| DCSwin | 95.93 | 92.72 | 83.37 | 88.94 | 77.64 | 87.72 | 91.76 | 78.73 |
| $A^2$-FPN | 96.56 | 94.70 | 84.15 | **89.68** | 83.11 | 89.64 | 92.78 | 81.66 |
| MANet | 96.14 | 93.77 | 82.95 | 89.01 | 82.87 | 88.95 | 92.09 | 80.53 |
| MAResUNet | 96.41 | 94.21 | 84.15 | 89.59 | 83.72 | 89.62 | 92.59 | 81.58 |
| DGCFNet | **96.74** | **94.75** | **84.39** | 89.58 | **84.50** | **89.99** | **92.90** | **82.20** |

Note:
Values highlighted in bold are the top performance.

**Table 5 Quantitative comparison with state-of-the-art models on the Potsdam dataset (%).**

| Method | Imp.surf. | Building | Lowveg. | Tree | Car | Mean F1 | OA | mIoU |
|---|---|---|---|---|---|---|---|---|
| FCN | 91.87 | 93.90 | 84.41 | 85.09 | 91.04 | 89.26 | 88.16 | 80.82 |
| BANet | 90.93 | 91.85 | 84.45 | 84.13 | 92.83 | 88.84 | 87.31 | 80.12 |
| LANet | 92.67 | 94.69 | 85.85 | 86.49 | 93.97 | 90.74 | 89.45 | 83.26 |
| DCSwin | 91.57 | 93.60 | 83.76 | 82.47 | 93.33 | 88.95 | 87.68 | 80.43 |
| $A^2$-FPN | 92.73 | 94.97 | 85.86 | 86.59 | 94.06 | 90.84 | 89.54 | 83.45 |
| MANet | 91.77 | 93.64 | 85.00 | 85.39 | 94.10 | 89.98 | 88.42 | 82.02 |
| MAResUNet | 92.45 | 94.82 | 85.84 | 86.38 | **94.49** | 90.80 | 89.45 | 83.38 |
| DGCFNet | **92.90** | **95.35** | **86.33** | **86.67** | 94.14 | **91.08** | **89.80** | **83.84** |

Note:
Values highlighted in bold are the top performance.

increases by 2.20%. Meanwhile, the proposed method outperforms BANet in all categories, especially in the small target category Car, achieving an F1-score of 84.50%, which is 4.32% higher than BANet. The above results consistently prove that the proposed method not only has stronger global contextual modeling ability, but also has significant improvement in the segmentation of small targets.

Compared with other models, the proposed method effectively reduces segmentation errors, and has better segmentation performance at different scales. For example, in the large target category of Building, our method can more accurately segment the complete boundary of objects. The above results show that using a hybrid model of CNN and Transformer is feasible and can produce even better results.

(2) Results on Potsdam dataset: The quantitative comparison results of different methods on the Potsdam dataset are shown in Table 5. It can be seen that the method proposed in this article achieves the highest mean F1, OA, and mIoU, and only slightly lower F1 values than the MAResU-Net method in the small target category of Car. Compared with BANet, the proposed method improves the value of mean F1 by 2.24%, OA by 2.49%, and mIoU by 3.72%. Especially in the large target category of Building, the F1 value is 3.50% higher than BANet, indicating that using a CNN and Transformer hybrid

**Table 6 Quantitative comparison with state-of-the-art models on the BLU dataset.**

| Method | Background | Built-up | Vegetation | Water | Agricultural | Road | Mean F1 | OA | mIoU |
|---|---|---|---|---|---|---|---|---|---|
| FCN | 64.60 | 80.85 | 88.22 | 78.35 | 86.29 | 57.81 | 76.02 | 83.44 | 62.57 |
| BANet | 69.00 | 85.58 | 89.81 | 77.05 | 85.49 | 60.05 | 77.83 | 85.06 | 64.87 |
| LANet | 71.83 | **87.28** | **90.36** | 79.01 | 86.47 | **69.02** | 80.66 | **86.31** | 68.34 |
| DCSwin | 68.79 | 84.02 | 89.10 | 78.61 | 83.68 | 66.52 | 78.42 | 84.18 | 65.25 |
| A²-FPN | **72.66** | 86.26 | 90.25 | 79.09 | 86.61 | 67.53 | 80.40 | 86.17 | 67.98 |
| MANet | 70.83 | 84.95 | 89.29 | 75.61 | 82.83 | 66.88 | 78.40 | 84.38 | 65.17 |
| MAResUNet | 71.12 | 85.99 | 90.28 | 76.60 | **87.16** | 67.54 | 79.78 | 86.11 | 67.20 |
| DGCFNet | 71.25 | 86.68 | 90.32 | **83.08** | 86.39 | 68.35 | **81.01** | 86.12 | **68.87** |

**Note:**
All scores are expressed as percentages (%). Values highlighted in bold are the top performance.

model for spatial feature extraction in large-scale remote sensing images has a significant advantage in semantic understanding ability.

The edges of the "clutter/background" and "Building" classes have similar colors and are adjacent to the Imperfect Surface class. Therefore, the edges of the Building class are easily misclassified as the "clutter/background" class. The proposed method can help the model better cope with noise and interference in images by fusing feature maps at different levels, and obtain a rich global contextual relationship, enabling the model to recognize semantic information and suppress the influence of noise. Similarly, the small target category of Imperfect Surface is located near the Low Vegetation and Building, which can easily cause segmentation errors. Our model can identify the above regions well.

(3) Results on BLU dataset: The quantitative results of different methods on the BLU dataset are shown in Table 6. The results indicate that our method achieves the highest meanF1 and mIoU values. Moreover, in the Water category, the proposed method can achieve the highest F1 score, reaching 83.08%. Compared with BANet, the meanF1 of DGCFNet method increases by 3.18%, OA increases by 1.06%, and mIoU increases by 4.00%. Especially on the large targets such as Water, our method outperforms BANet by 6.03%, indicating that it can more comprehensively utilize the contextual information of images. However, other categories did not achieve the best results, except for the Water category. Due to its large area and relatively distinct characteristics, the Water category achieved good results that are significantly better than those of other categories. While the results for the other categories are slightly lower, this does not affect the final average result.

Since the Water class and the adjacent Farmland class share similar colors, other methods may mistakenly segment Water as Farmland due to insufficient discriminative information. Additionally, other methods are prone to significant omissions due to the limitations of their receptive fields. However, compared to other methods, our model, which introduces a hybrid structure of CNN and Transformer, achieves more complete and smoother boundary segmentation.

**Table 7 Quantitative comparison results on the Potsdam dataset with state-of-the-art networks.**

| Method | FCN | BANet | LANet | DCSwin | MANet | DGCFNet |
|---|---|---|---|---|---|---|
| Parameters (M) | 11.3 | 12.7 | 23.8 | 45.6 | 35.9 | 23.9 |
| Memory (MB) | 172.0 | 194.6 | 363.8 | 694.6 | 574.9 | 280.7 |
| mIoU (%) | 80.8 | 80.1 | 83.3 | 80.4 | 82.0 | 83.8 |

Note:
The parameters and memory are measured using a 512 × 512 input on a single NVIDIA Tesla A800 GPU.

In the evaluation of network models, parameters and memory are crucial for assessing the network. We compared our DGCFNet with other segmentation networks on the Potsdam dataset, as depicted in Table 7.

## Limitation

Although the proposed DGCFNet has better understood the relationship between different scales and semantic features by utilizing intra-layer and inter-layer information, there are still potential aspects that need to be considered.

Firstly, the total number of parameters in DGCFNet is 23.9M, which is smaller than that of medium-sized networks like DCSwin (45.6M) and MANet (35.9M), but larger than that of small-sized networks such as FCN (11.3M) and BANet (12.7M). Although DGCFNet has fewer parameters compared to DCSwin and MANet, it achieves a better IoU. However, while it achieves a higher mIoU than FCN and BANet, it consumes more computational resources. This indicates that our network structure does well in balancing accuracy and efficiency, but there is still room for reducing the model size in the decoder by employing model pruning techniques.

Secondly, although we have improved the segmentation capability for targets of different sizes by incorporating a global compensation branch in the Double-branch Global Extraction Module, there is still significant room for improvement in segmenting targets within complex backgrounds. Therefore, our future work will focus on model lightweighting and enhancing the capability to segment small targets.

## CONCLUSION

This article proposes a remote sensing image semantic segmentation network DGCFNet based on a serial structure of CNN and Transformer. The DGCFNet network extracts local features through CNN and embeds DGEM into the Transformer to obtain global contextual relationships. Meanwhile, different levels of contextual information are fused through CIIM. In addition, the FIGM is proposed to adaptively fuse the intra-layer contextual information in DGEM with the inter-layer contextual information in CIIM. Extensive experiments conducted on three high-resolution remote sensing datasets (Vaihingen, Potsdam and BLU) demonstrate the reliability and effectiveness of the proposed method in remote sensing image semantic segmentation tasks.

Although the proposed method, DGCFNet, can accurately segment multiple targets in remote sensing images, our focus on improving long-range contextual relationships has

significantly enhanced the segmentation of larger objects. However, this emphasis has inadvertently led to the neglect of segmenting small targets in complex backgrounds, primarily manifesting as an inability to fully fit the targets and resulting in boundary segmentation errors. In the future, we will represent the similarity of objects within the same category by calculating a mutual relationship matrix between intermediate feature blocks and image patches. This relationship will be utilized for the segmentation of small targets.

### Funding
This research was funded by the Key Research and development projects in Henan province (Grant No. 241111210100), Key scientific research projects of colleges and universities in Henan province (Grant No. 23A520028, 24B120009), and Zhongyuan University of Technology Strength Enhancement Program for Advantageous Subjects (Grant No. GG202415). The funders had no role in study design, data collection and analysis, decision to publish, or preparation of the manuscript.

### Grant Disclosures
The following grant information was disclosed by the authors:
Key Research and Development Projects in Henan Province: 241111210100.
Key Scientific Research Projects of Colleges and Universities in Henan Province: 23A520028, 24B120009.
Zhongyuan University of Technology Strength Enhancement Program for Advantageous Subjects: GG202415.

### Competing Interests
The authors declare that they have no competing interests.

### Author Contributions
- Yuan Liao conceived and designed the experiments, performed the experiments, performed the computation work, prepared figures and/or tables, authored or reviewed drafts of the article, and approved the final draft.
- Tongchi Zhou performed the experiments, authored or reviewed drafts of the article, and approved the final draft.
- Lu Li analyzed the data, prepared figures and/or tables, authored or reviewed drafts of the article, and approved the final draft.
- Jinming Li analyzed the data, prepared figures and/or tables, authored or reviewed drafts of the article, and approved the final draft.
- Jiuhao Shen analyzed the data, prepared figures and/or tables, authored or reviewed drafts of the article, and approved the final draft.
- Askar Hamdulla performed the experiments, authored or reviewed drafts of the article, and approved the final draft.

## Data Availability

The code is available at GitHub and Zenodo:

- https://github.com/liaoyuan0604/DGCFNet.

- Liao,Yuan. (2025). DGCFNet [Data set]. Zenodo. https://doi.org/10.5281/zenodo.14614699.

The code is written in Python and can be opened using the official PyCharm software: https://www.jetbrains.com/pycharm.

The Vaihingen dataset and the Potsdam dataset are aerial image datasets produced by the German Aerospace Center-DLR available at: https://www.isprs.org/education/benchmarks/UrbanSemLab/Default.aspx.

The BLU dataset, an aerial image dataset produced by the Beijing-2 satellite provided by the 21st Century Aerospace Technology Company Ltd, is available at: https://rslab.disi.unitn.it/dataset/BLU.

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
