# Peer review of "DGCFNet: Dual Global Context Fusion Network for remote sensing image semantic segmentation"

_PeerJ Computer Science, doi:10.7717/peerj-cs.2786_

## Round 0.1 · original submission · Major Revisions

The manuscript presents a novel approach to remote sensing image semantic segmentation that integrates CNNs and Transformers, leveraging several custom-designed modules to enhance feature correlation, intra- and inter-layer information fusion, and overall segmentation performance. The reviewers commend the scientific novelty, clarity of organization, and comprehensive experimentation. However, several issues raised by reviewers must be addressed before the manuscript is deemed suitable for publication.

Reviewer 1 ·

Basic reporting

no comment

Experimental design

no comment

Validity of the findings

no comment

Additional comments

This manuscript proposed a novel remote sensing image semantic segmentation network through effectively combining CNN and Transformer, where many smart modules were further designed to enhance the ability of Transformer, the correlation between features at different levels, and the fusion of intra- and inter-layer information. The extensive experiments were conducted to validate the proposed method. Overall, the organization and writing of this manuscript is well and easy to follow, and the scientific novelty is strong, which makes the manuscript good enought to be published after minor revisions.

1. in eq. (11), "(.) represents the Sigmoid activation function" is confusing. Does the authors want to state "Sigmoid (.) represents the Sigmoid activation function"? If so, what does the operation of "." in the Sigmoid mean?

2. The experiments were conducted on three datasets, i.e., Vaihingen, Potsdam, and BLU. Are they public datasets? If so, suggest to add some kinds of references for them.

3. In the subsection of "Comarison with other methods", suggest to briefly introduce the parameter settings of the competing methods.

Reviewer 2 ·

Basic reporting

There are several grammatical errors and awkward phrasings that affect readability. A thorough proofreading is recommended.
e.g. Syntax: L18: ... models need to have both rich local ... suggest revising to ... models need to capture both rich local ...
Capitalize: L192;L199 Capitalize the initial letter of 'transformer' to 'Transformer'. The title may need as well in some words.
Formatting: Reference e.g. L158: Transformer(Vaswani, 2017) L143:DeeplabV3(Chen, 2017) VS L73:Transformer (Vaswani, 2017) L150:LANet (Ding et al., 2020)
Obvious mistakes: L268: mean intersection ratio sum (mIoU) is not right, it should be mean intersection over union (or Jaccard index)

The literature references need updating, as there are only two from 2023 and none from 2024, despite being published at the end of 2024.
When mentioning previous works, briefly describe their key contributions and limitations and how these methods influence your work.

Improve the figure resolutions, all the figure is lossy compressed; and increase the font size in the diagram. (Figure 2)

The future work in conclusion is too brief and strange: Please read??? L370-L372:The next step will be to explore methods to improve the model’s ability to segment small objects. The future work will explore ways to improve the ability to segment small targets.

Qualitative analysis expression need imporvement and clearify. L353: From the first row, it can be seen that Water class and the adjacent Farmland class have similar colors.
Is it the input image NOT first row? Otherwise, it doesn't make sense, as GT is in the first row.

Missing limitation and error analysis.

Strengths: The manuscript presents a cohesive study, including the motivation, proposed method, experimental setup. Ablation studies and comparisons with other methods(while not latest) are provided to validate the effectiveness of the proposed network.

Experimental design

Address the potential weaknesses of the proposed method, including any constraints in its application or scenarios where its performance may be suboptimal.
Compare the computational cost and the number of parameters required by the method against alternative approaches to highlight trade-offs.
If feasible, commit to sharing the codebase or providing access to it upon publication to promote transparency and reproducibility.

Validity of the findings

The experiments were conducted on three widely recognized datasets: Vaihingen, Potsdam, and BLU. Evaluation metrics included Overall Accuracy (OA), Mean Intersection over Union (mIoU), and Mean F1 Score (mF1), which are standard and well-suited for semantic segmentation tasks. Ablation studies were performed to assess the contribution of each component of the proposed network. The proposed DGCFNet demonstrated slightly better segmentation performance compared to several existing methods (until 2023) across the Vaihingen, Potsdam, and BLU datasets. However, the qualitative analysis requires significant improvement to better convey the results and insights.

Finally, providing access to the datasets and code, if feasible, would greatly enhance the reproducibility and impact of this work.

Additional comments

A complete proofreading is necessary before submit to review.

·

Basic reporting

The authors have proposed a meaningful and effective remote sensing image semantic segmentation approach, which captures both local detailed information and long-range context information by fusing CNN and Transformer. Specifically, three key components, dual-branch global extraction module (DGEM), cross-level information interaction module (CIIM), and feature interaction guided module (FIGM) are incorporated to achieve robust and accurate semantic segmentation.

Experimental design

Extensive experiments conducted on three semantic segmentation datasets demonstrate the effectiveness of the proposed model and its superiority over state-of-the-arts.

Validity of the findings

have proposed a meaningful and effective remote sensing image semantic segmentation approach, which captures both local detailed information and long-range context information by fusing CNN and Transformer.

Additional comments

1. According to the authors’ opinion, existing combining CNN and Transformer” methods only use Transformer in the encoder, there is little research in the decoder. It is suggested to add some explanations of the rationality and feasibility of applying the Transformer in the decoder.
2. In Figure 1, what are the objects encompassed by red, yellow, and purple boxes? It is hard to distinguish.
3. The three datasets in the experiments should be cited.
4. From Table 1 to Table 3, some categories’ F1 scores decrease when adding the designed modules, such as Imp. Surf and Building in Table 1, Imp. Surf and Car in Table 2, and Tree and Car in Table 3. It is suggested to add some discussion for these results.
5. Typos.
In the abstract, To further enhance…  to
On page 3, WiCoNet to extracts …and then fuses… extract...fuse
It uses transposed convolution layers  fully
On page 4, LANet …utilize multi-scale feature fusion  utilizes
SETR utilizes the self-attention…  utilized
In Page 5, DGEM module, CIIM module, FIGM module  DGE module, CII module, FIG module.

Please proofread the whole paper and double check similar typos, and revise them.

---

## Round 0.2 · Minor Revisions

The authors have addressed major concerns raised by the reviewers. I encourage the authors to address the minor amendments further before the manuscript can be accepted.

Reviewer 1 ·

Basic reporting

no comment

Experimental design

no comment

Validity of the findings

no comment

Additional comments

The authors have well solved my concerns. The current version of this manuscript is ready for publication.

·

Basic reporting

The authors have updated the manuscript according to the comments raised before. However, there are still some issues that need to be refined before the final publication. The comments are listed as follows.

1. Figure 1 lacks input and output, making the reader cannot make sense.
2. In the related work, it is suggested that the difference between the existing work and the proposed model be highlighted.
3. In Table 1, the table appears unattractive after using line breaks. It is recommended not to use newlines.
4. In Table 6, for the F1 scores for each category, only "water" achieves the best result, while others not. Please analyze the reason.

Experimental design

no comment

Validity of the findings

no comment

---

## Round 0.3 · accepted · Accept

The authors have addressed all of the reviewers' comments. The manuscript is ready for publication. I'm happy to accept it.